# Factors affecting spatio-temporal occurrence of sympatric civets in Parsa-Koshi Complex, Nepal

**Bishal Subedi**[1,2], **Sandeep Regmi**[1,2,3,4], **Amrit Nepali**[1,2], **Niraj Regmi**[1], **Amir Basnet**[1], **Krishna Tamang**[1], **Bishnu Aryal**[1,2], **Sabin KC**[1], **Pradip Kandel**[1], **Shivish Bhandari**[5], **Bishnu Prasad Bhattarai**[1,4], **Hem Bahadur Katuwal**[3,4], **Jerrold L. Belant**[6], **Ashok Kumar Ram**[7], **Hari Prasad Sharma**[1,4]*

1 Central Department of Zoology, Institute of Science and Technology, Tribhuvan University, Kirtipur, Kathmandu, Nepal, 2 The Himalayan Conservancy, Kirtipur, Kathmandu, Nepal, 3 Southeast Asia Biodiversity Research Institute, Chinese Academy of Sciences and Center for Integrative Conservation, Xishuangbanna Tropical Botanical Garden, Chinese Academy of Sciences, Mengla, Yunnan, China, 4 Nepal Zoological Society, Kirtipur, Kathmandu, Nepal, 5 Department of Biology, Morgan State University, Baltimore, Maryland, United States of America, 6 Department of Fisheries and Wildlife, Michigan State University, East Lansing, Michigan, United States of America, 7 Department of National Parks and Wildlife Conservation, Kathmandu, Nepal

* hari.sharma@cdz.tu.edu.np

## Abstract

Understanding the effect of biotic and abiotic factors, including habitat and interspecific competition, is crucial for species conservation. We quantified spatio-temporal patterns of sympatric large Indian civet (LIC; *Viverra zibetha*) and small Indian civet (SIC; *Viverricula indica*) using remote cameras in Parsa-Koshi Complex, Nepal during December 2022–March 2023. We found low spatial overlap between LIC and SIC ($O_{ij} = 0.15$) and high diel overlap between LIC and SIC ($D_{hat1} = 0.759$, normo0 CI: $0.670 - 0.847$). Large predators, i.e., tigers (*Panthera tigris*) and leopards (*P. pardus*) positively influenced the occurrence of LIC and SIC, respectively. Extent of grassland also positively influenced ($0.529 \pm 0.193$) SIC occurrence. The coexistence of LIC and SIC is governed by complex ecological interactions, including habitat preferences and the influence of predator's occurrences, and such dynamics are important implications for conservation planning. Effective conservation strategies should be considering for the spatial and temporal overlap of these species, considering the role of large predators and habitat variables such as grasslands to support the coexistence of sympatric carnivores and reduce human-wildlife conflict.

## Introduction

Mammalian species interact with their environment and each other, often mediated by biotic and abiotic factors that influence their ecology [1,2]. These interactions play a crucial role in shaping community dynamics and ecosystem functioning [3,4]. Theories such as niche differentiation and resources partitioning suggest that sympatric

**Data availability statement:** The data are available fin the manuscript.

**Funding:** Tribhuvan University National Priority Area Research Grant (TU-NPAR-2078/79-ERG-04) of Tribhuvan University, Nepal.

**Competing interests:** The authors have declared that no competing interests exist.

species can coexist by minimizing competition through spatial, temporal and dietary adjustments [5–7]. These mechanisms allow species to occupy distinct ecological niches, facilitating biodiversity and ecosystem stability [8–10].

Small carnivores (i.e., < 16 kg body mass) [11], due to their intermediate trophic position, are particularly important for understanding these dynamics [12]. They influence ecosystems by controlling prey populations, aiding seeding dispersal, and maintaining ecological balance [13]. Study have shown that sympatric small carnivores often exhibit niche partitioning to reduce competition [14,15]. For example, jungle cats (*Felis chaus*) and leopard cats (*Prionailurus bengalensis*) in Nepal demonstrated temporal adjustments [16], while sympatric civet species in Pakistan partitioned habitat use to minimize overlap, whereas Asian palm civet (*Paradoxurnus hermaphroditus*) and small Indian civet (SIC; *Viverricula indica*) altered habitat use by being active immediately after following peak activity of other civets [17] Additionally, interactions with higher trophic levels, such as large predators, can influence small carnivore activity and distribution patterns, as observed in Huai Kha Khaeng Wildlife Sanctuary, Thailand [18].

Civets are widely distributed nocturnal small carnivores inhabiting diverse habitats across southern and southeast Asia [19–22]. Among them, both the large Indian civet (LIC; *Viverra zibetha*) and SIC co-occur in several regions [19,20]. These species exhibit ecological overlap in terms of diet and habitat use, but factor like forest type, elevation, presence of large predators, and human activity mediate their occurrence [19]. The LIC prefers dense forests and elevations up to 3080 m [20,22], whereas the SIC is associated with more open habitats and elevations up to 2500 m [20,21]. Both species are known to coexist with large predators, suggesting a dynamics interplay of competition and predation in structuring their ecological niches [23]. Different habitat selection of civet species to occupy different ecological niches leading to spatial segregation to reduce competition [24].

LIC and SIC often exhibit niche differentiation [19], reportedly to reduce competition and facilitate co-occurrence [25]. The SIC primarily consumes termites, rodents, fruits and poultry, while the LIC feeds on broader range of prey, including rodents, reptiles and birds [20]. Such dietary and spatial segregation likely underpins their coexistence. However, these civets face threats such as habitat loss, poaching and human-wildlife conflict, which may influence their behavior and distribution [20,26].

Despite their ecological importance, limited information is available regarding the spatial and temporal distribution of LIC and SIC in Nepal. To address these gaps, our study aimed to characterize factors influencing the occurrence of these two civet species in the Parsa-Koshi Complex (PKC), Madhesh Province, Nepal. Specifically, we investigated how these sympatric species partitioning resources spatially and temporally in relation to environmental factors, human activity, and the presence of large predators. Based on niche theory and prior evidence, we assumed that LIC and SIC would coexist within shared habitats, exhibiting reduced dial overlap in areas with high predator activity. This study contributes to a better understanding of carnivore interactions and their implications for biodiversity conservation, particularly in a landscape where human-wildlife interactions are increasingly pronounced.

## Materials and methods

### Study area

We conducted this study within Parsa-Koshi Complex (PKC), Madhesh Province, Nepal, encompassing the area between Parsa National Park (PNP) in the west and Koshi Tappu Wildlife Reserve (KTWR) in the east (Fig 1). This area is situated in lowland Nepal and comprises 9661 km2 with an elevation range of 80–910 m. The PKC primarily contains sub-tropical forests, with sal (*Shorea robusta*) and mixed forests dominated by acacia (*Acacia catechu*) species [27, 28, 29, 30].

In addition to LIC and SIC, mammal species in PKC include tiger, common leopard, striped hyaena (*Hyaena hyaena*), Bengal fox (*Vulpes bengalensis*), golden jackal (*Canis aureus*), jungle cat, barking deer (*Muntiacus vaginalis*), wild boar (*Sus scrofa*), sloth bear (*Melursus ursinus*) and Asian elephant (*Elephas maximus*) [20,21,27,31]. Human communities living in the PKC area depend on crop and livestock agriculture for their livelihoods [27,30]. Forest products including fire-wood, leaves, and wood are harvested for subsistence [30].

### Data collection

We collected LIC and SIC occurrence data during December 2022–March 2023. For data collection including camera placement and camera setup, we followed [31]. We established a 5 km x 5 km grid and systematically deployed four cameras (Stealth Cam STCG45NG) in each cell separated by at least 1 km (154 cameras total). We emphasized the placement of cameras on core forest and edge habitats, while core human settlement area was excluded. We chose these grids location based on their accessibility. We positioned cameras 40–60 cm above ground along trails and paths used by wildlife. At each placement, we inspected the area for wildlife signs such as tracks or scats, adjusted the angle and height of optimize coverage, and conducted test shots to confirm the field of view to address the risk of missed detections. We programmed each camera to take three images each detection, with a 30-sec delay between detections. Cameras were

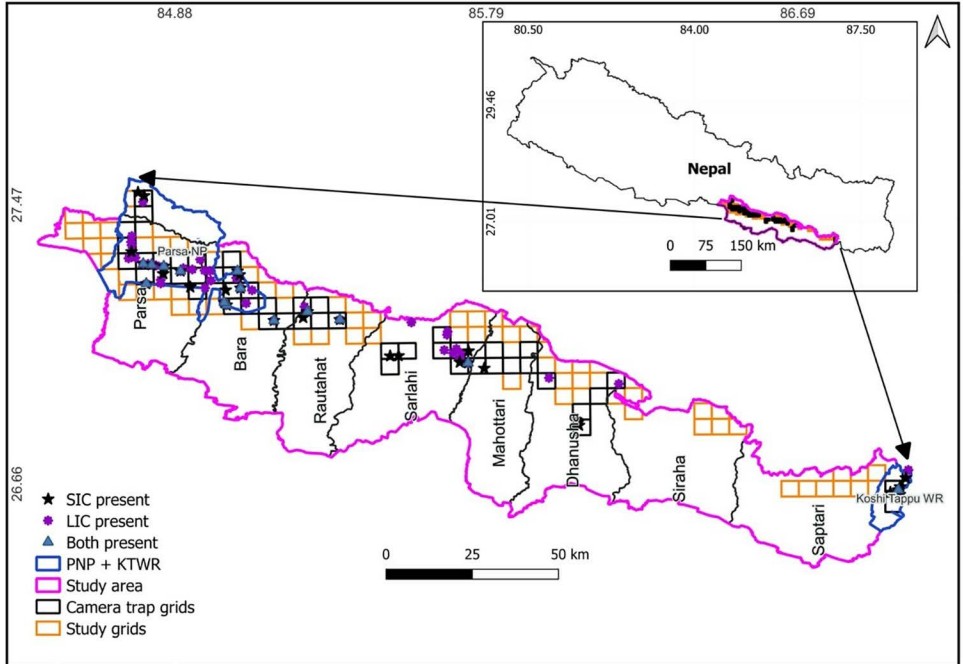

**Fig 1. Study area of large Indian civet and small Indian civet, Parsa- Koshi Complex, Nepal, December 2022–March 2023.** Contains information from OpenStreetMap and OpenStreetMap Foundation, which is made available under the Open Database License [35].

operational for 21 days. Due to relatively higher spatial extent of PKC than available cameras, the cameras were shifted every 21 days after the sampling is completed.

At each camera station, we recorded habitat variables such as canopy cover and distance to the nearest permanent water body and human settlement in the field. Further, we also took the area of grassland within 500 m radius of trap points using ESRI Sentinel-2 land-use land-cover map 2022 at a 10 m resolution [32]. We established a 10 m x 10 m plot at each camera station, keeping the camera as the center, and we estimated canopy cover as average value from the four corners and plot center using the Gap Light Analysis mobile application [GLAMA; 33]. We obtained the number of humans and large predators detected from each camera. We measured distance to nearest settlement, nearest water body, and nearest road using measuring tape when ≤200 m from each camera and QGIS [34] when distances exceeded 200 m. We extracted the data on district boundary of the study area from the Nepal administrative boundaries shapefile downloaded from the website of Humanitarian Data Exchange (https://data.humdata.org/dataset/cod-ab-npl). Similarly, the boundary files of protected areas were downloaded from the Open Street MaP [35] and mapped in QGIS [34].

### Ethical considerations for camera trap studies

Camera traps research permission was obtained from the Department of Forest and Soil Conservation (Permission Number: 596) and the Department of National Parks and Wildlife Conservation (Permission Number: 1165). We informed peoples of our use of remote camera before deploying.

### Data analysis

We assessed correlations between continuous variables using a threshold of $|r| > 0.7$ [36], and none were highly correlated (S1 Fig). We used a Generalized Linear Model (GLM) with a binary response variable (presence/absence) to identify factors affecting the occurrence of LIC and SIC [37]. The covariates included in the models were selected based on ecological relevance and prior studies (e.g., area of habitat type, such as farmland, and grassland, percent canopy cover, human presence, presence of large predators such as tiger and leopard, distance to settlements and waterbodies). Model selection was performed using all possible combinations of explanatory variables, and the best fitting model were ranked according to $\Delta AICc \leq 2$. We conducted model averaging based on best supported models to obtain parameter estimates and their association uncertainties [38]. All analysis was performed in R program using the packages wiqid, MuMIn, overlap, tidyverse, corrplot [39].

We recorded the site, date, and the time of detection from camera images for LIC and SIC, considering detections of the same species within 30 minutes as a single event to facilitate independence. We estimate Pianka's niche overlap index by adding up the frequency of detections for each species at each site [40]. This index was chosen due to its simplicity and adaptability in measuring resources use overlap between two species. The values of Pianka's index range from zero indicating no overlap to one indicating complete overlap [40], derived using the equation:

$$Ojk = \frac{\sum_i^n PijPik}{\sqrt{\sum_i^n P^2ij \sum_i^n P^2jk}}$$

where *pij* and *pik* denotes the relative frequency of images at site *l* for species *j* or *k*.

We estimated temporal overlap between LIC and SIC using the package Overlap [41] in R program [39]. We converted the time of each capture into radians to generate circular distributions of temporal data [42] and calculated activity overlap coefficients (Dhat1) for each species. The area under the density curves represents the coefficient of overlap, degree of overlap using circular kernel density estimates by incorporating the minimum density function from the two samples of detection data compared at each point in a 24-hour period [43]. We categorized diel activity patterns on LIC and SIC

based on the proportion of time spent engaged in activities occurring at night. Species were classified as nocturnal if 70% or more of their activity took place during nighttime, cathemeral if 30–70% occurred at night, and diurnal if 10–30% of their activity was nocturnal. Crepuscular behavior was defined when 50% of the activity occurred within one hour before or after sunrise or sunset [44–46]. We used 999 bootstraps to generate 95% confidence intervals [47]. The intensity of temporal overlap was denoted by the coefficient of overlap, with zero denoting no overlap and one denoting complete overlap [43,47].

## Results

In 3234 total camera nights, we detected LIC at 46 sites (112 detections), SIC at 36 sites (71 detections), and both species at 14 of 154 sites. The average probability of large carnivore detection and human detection was 0.357 ± 0.481, and 63.1 ± 237.9, respectively. Average canopy cover was 40.8 ± 22.2% whereas the average distance to the nearest waterbody was 2196 ± 2222 m, and average distance to the nearest human settlement was 3211 ± 1932 m. Average area of farmland and grassland was 0.052 ± 0.021 km$^2$ and 0.021 ± 0.051 km$^2$, respectively.

The best supported model for SIC included farmland area, grassland area and presence of large predators (AICc weight = 0.061) (Table 1), and the best supported model for LIC included large predators (AICc weight = 0.053) (Table 2). Occurrence of SIC increased with increasing large predator presence (0.454 ± 0.229; Estimate ± SE) followed by the grassland area (0.529 ± 0.193) (Fig 2; Table 3). Similarly, LIC occurrence increased with large predator presence (0.777 ± 0.199) (Fig 3). No other factors influenced occurrence of LIC or SIC (Table 3).

We identified low spatial overlap between LIC and SIC (Oij = 0.15) and high diel overlap between LIC and SIC (Dhat1 = 0.759, normo0 CI: 0.670 – 0.847) (Fig 4). We found LIC was most active during 1:00–2:00 and 8:00–9:00 hrs and SIC was most active during 2:00–3:00 hrs, followed by 7:00–10:00 (Fig 4).

Both the species demonstrated diurnal to crepuscular behaviour and had similar diel overlap (Dhat1 = 0.653, norm0 CI: 0.461–0.845) between LIC and SIC detected at the same sites.

**Table 1. Logistic Regression Model describing the factors affecting Small Indian Civet (SIC) Occurrence during 2022—2023, ranked according to the Akaike Information Criterion adjusted for small sample size (AICc).** Model parameters include SIC (presence absence) as response variable while Large Indian Civet (LIC; presence absence), farmland area (km$^2$), grassland area (km$^2$), canopy cover (%), presence of large predators, number of presence human, nearest distance to settlement (m), nearest distance to permanent water bodies (m) as predictive variables. K is the number of parameters, ΔAICc is the difference between the AICc value of the best-supported model and successive models, and Wi is the Akaike model weight.

| S.N. | Covariates | k | AICc | ΔAICc | wi |
|---|---|---|---|---|---|
| 1 | Farmland Area + Grassland Area + Presence of Large Predators | 4 | 161.805 | 0.000 | 0.061 |
| 2 | Farmland Area + Grassland Area + Presence of Large Predators + Distance to Water | 5 | 162.951 | 1.146 | 0.035 |
| 3 | LIC + Farmland Area + Grassland Area + Presence of Large Predators | 5 | 163.039 | 1.233 | 0.033 |
| 4 | Grassland Area + Large Predators | 3 | 163.207 | 1.401 | 0.030 |
| 5 | Distance to Settlements + Farmland Area + Grassland Area + Presence of Large Predators + Distance to Water | 6 | 163.449 | 1.644 | 0.027 |
| 6 | Canopy Cover + Farmland Area + Grassland Area + Presence of Large Predators | 5 | 163.527 | 1.722 | 0.026 |
| 7 | Distance to Settlements + Farmland Area + Grassland Area + Presence of Large Predators | 5 | 163.808 | 2.003 | 0.023 |
| 8 | Farmland Area + Grassland Area + Presence Number of Human + Presence of Large Predators | 5 | 163.895 | 2.090 | 0.022 |
| 9 | LIC + Farmland Area + Grassland Area | 4 | 164.131 | 2.326 | 0.019 |
| 10 | Null | 1 | 169.512 | 7.707 | <0.001 |

**Table 2. Logistic Regression Model describing the factors affecting Large Indian Civet (LIC) Occurrence during 2022—2023, ranked according to the Akaike Information Criterion adjusted for small sample size (AICc).** Model parameters include LIC (presence absence) as response variable while Small Indian Civer (SIC; presence absence), farmland area (km²), grassland area (km²), canopy cover (%), presence of large predators, number of presence human, nearest distance to settlement (m), nearest distance to permanent water bodies (m) as predictive variables. K is the number of parameters, ΔAICc is the difference between the AICc value of the best-supported model and successive models, and Wi is the Akaike model weight.

| S.N. | Covariates | k | AICc | ΔAICc | wi |
|---|---|---|---|---|---|
| 1 | Presence of Large Predators | 2 | 174.187 | 0.000 | 0.053 |
| 2 | Distance to Settlements + Presence of Large Predators | 3 | 175.187 | 1.000 | 0.032 |
| 3 | Number of Human Presence + Presence of Large Predators | 3 | 175.188 | 1.001 | 0.032 |
| 4 | SIC + Presence of Large Predators | 3 | 175.342 | 1.155 | 0.030 |
| 5 | Canopy Cover + Presence of Large Predators | 3 | 175.406 | 1.219 | 0.029 |
| 6 | Distance to Settlements + Number of Human Presence + Presence of Large Predators | 4 | 175.724 | 1.537 | 0.025 |
| 7 | Farmland Area + Presence of Large Predators | 3 | 175.762 | 1.575 | 0.024 |
| 8 | Grassland Area + Presence of Large Predators | 3 | 175.876 | 1.689 | 0.023 |
| 9 | Presence of Large Predators + Distance to water | 3 | 176.215 | 2.028 | 0.019 |
| 10 | Null | | 189.832 | 15.645 | <0.001 |

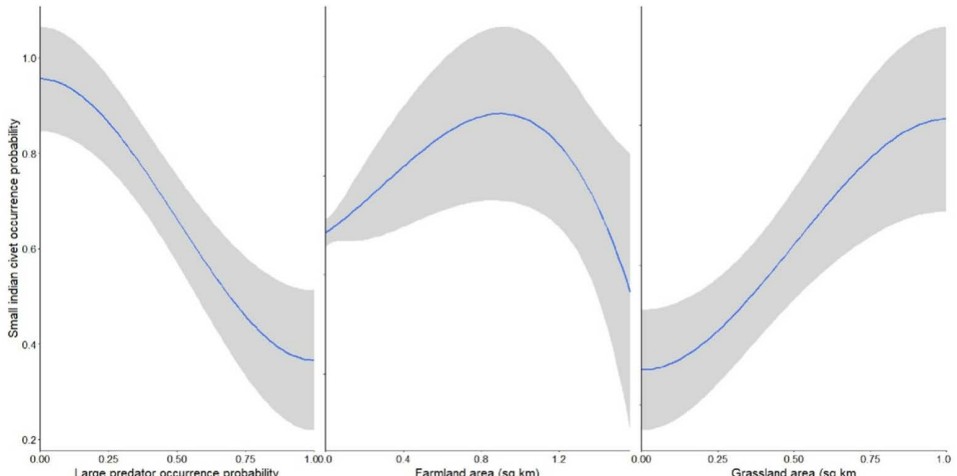

**Fig 2. Response plot of small Indian civet (SIC) occurrence probability with the top model of the variables.**

## Discussion

We found the presence of large predators and grassland, influenced the occurrence of LIC and SIC. These findings align with other studies that highlights the role of habitat structure and predator presence in shaping small-carnivore distribution and interactions. The low spatial overlap between LIC and SIC might be due to niche partitioning driven by habitat preferences and predator avoidance, a common mechanism facilitating coexistence in sympatric carnivores [48]. Spatial segregation between Asian palm civet and Small Indian civet, has also reported in fragmented forests of Pakistan [17]. The LIC mostly selects for dense forests, scrublands, and plantation forests whereas SIC occur in fragmented forests, and grasslands [20,24]. This Spatial partitioning minimizes direct competition and allows species to coexist despite overlapping geographical ranges. Such partitioning between smaller and meso-predators has been observed in multiple studies [16,17].

**Table 3. Model average parameter estimates and 95% Confidence Interval; lower confidence interval (LCI), upper confidence interval (UCI) described the presence of large Indian civet and small Indian civets in Parsa—Koshi Complex, Nepal, 2023. Model parameters includes in Table 1 and 2.**

**Large Indian civet**

| Parameters | Estimate | SE | LCI | UCI | z | p |
|---|---|---|---|---|---|---|
| Intercept | −0.996 | 0.210 | −1.413 | −0.579 | 4.685 | **<0.001** |
| Farmland Area | −0.255 | 0.342 | −0.933 | 0.422 | 0.739 | 0.459 |
| Grassland Area | 0.113 | 0.195 | −0.272 | 0.499 | 0.575 | 0.565 |
| Presence of Large Predator | 0.777 | 0.199 | 0.383 | 1.172 | 3.862 | **<0.001** |
| Distance to Water | 0.026 | 0.250 | −0.467 | 0.521 | 0.107 | 0.915 |
| Small Indian civet Detection | 0.427 | 0.434 | −0.430 | 1.285 | 0.977 | 0.328 |
| Distance to Settlement | −0.257 | 0.220 | −0.693 | 0.179 | 0.739 | 0.459 |
| Canopy Cover | −0.188 | 0.206 | −0.596 | 0.219 | 0.906 | 0.364 |
| Number of Human Presence | −0.263 | 0.335 | −0.926 | 0.400 | 0.778 | 0.436 |

**Small Indian civet**

| Parameters | Estimate | SE | LCI | UCI | z | P |
|---|---|---|---|---|---|---|
| Intercept | −1.337 | 0.238 | −1.807 | −0.867 | 5.576 | **<0.001** |
| Farmland Area | 0.341 | 0.178 | −0.012 | 0.695 | 1.891 | 0.058 |
| Grassland Area | 0.529 | 0.193 | 0.147 | 0.911 | 2.714 | **<0.001** |
| Presence of Large Predator | 0.454 | 0.229 | 0.001 | 0.907 | 1.964 | **<0.001** |
| Distance to Water | −0.323 | 0.279 | −0.875 | 0.228 | 1.151 | 0.249 |
| Large Indian civet Detection | 0.504 | 0.455 | −0.395 | 1.404 | 1.099 | 0.272 |
| Distance to Settlement | 0.250 | 0.295 | −0.332 | 0.833 | 0.843 | 0.399 |
| Canopy Cover | −0.161 | 0.228 | −0.612 | 0.290 | 0.701 | 0.483 |
| Number of Human Presence | −0.040 | 0.182 | −0.399 | 0.318 | 0.222 | 0.825 |

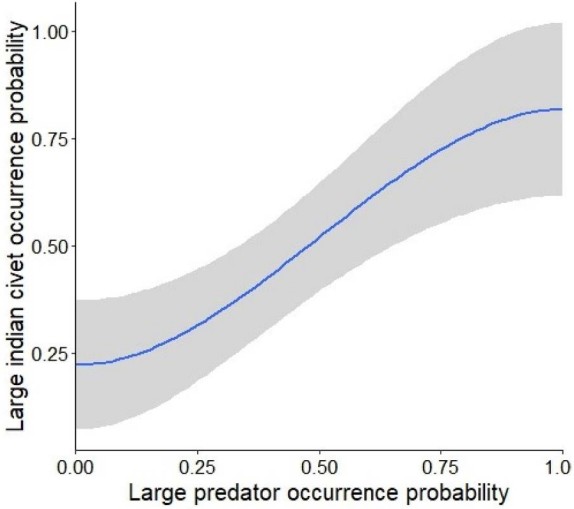

**Fig 3. Response plot of large Indian civet (LIC) occurrence probability with the top model of the variable.**

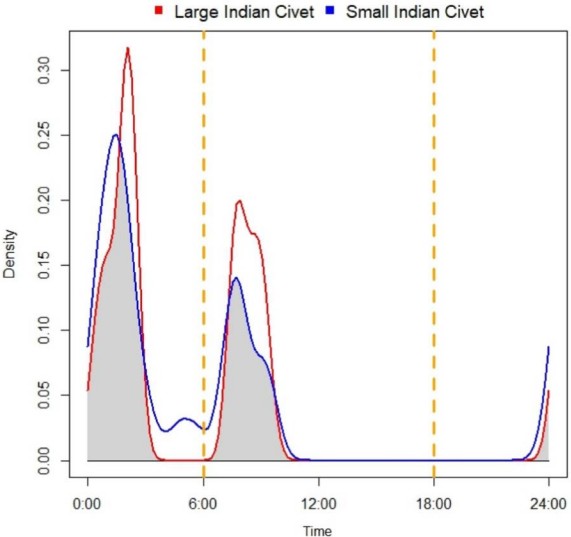

**Fig 4.** **Temporal overlap between large Indian civet (red line) and small Indian civet (blue line), Parsa-Koshi Complex, Nepal, 2022–2023.**

We observed the higher diel overlap between LIC and SIC indicating temporal convergence in their activity patterns. Both civet species exhibited greater activity after midnight until morning and were largely inactive from afternoon to midnight, which could be due to shared adaptations regarding hunting behaviour, prey availability, and avoidance of humans and large predators [49]. LIC and SIC have overlapping diets, which could result in these species sharing a similar niche due to activity of similar prey species [24].

The coexistence of LIC and SIC alongside large predators in the same habitat might be due to their potential scavenging of carrion remains from large predators [50,51]. Consequently, the presence of large predators can alter behaviour and habitat use of LIC and SIC, promoting co-occurrence through spatial and temporal niche divergence and spatial segregation among carnivore species [23].

We found that SIC occurrence increased with grassland area, which might be due to the availability of ground forage species [21,52,53]. For example, grasslands offer habitat for insects, lizards, small rodents, birds, snakes, and frogs [54,55], which are major food for civets [21,56].

Furthermore, we found no association of LIC and SIC with distance to nearest settlements, water body, and number of human presences. Generally, both LIC and SIC partitioned resource spatially, including grasslands and forests where both species occur and frequently hunt prey [21,22]. This could explain the lack of association of LIC and SIC with settlements, water bodies, farmland and human presence. Understanding the differences in habitat selection between LIC and SIC, where SIC select grassland areas and LIC select forested environments, has important implications for conservation management. This knowledge can help to develop conservation strategies that considers interactions with large carnivores, to address the ecological needs of each species. By aligning conservation interventions that emphasize observed habitat selection and considering their coexistence with large carnivores, we can also be used in human-wildlife conflict mitigation strategies to further conservation outcomes.

## Conclusions

The occurrence of sympatric LIC and SIC in PKC was influenced by grassland habitat, and large predators. Both species likely benefitted from abundant food with potential complex interactions in predator-prey dynamics. The occurrence and activity patterns of LIC and SIC suggest both are nocturnal and might consume similar foods, potentially causing

interspecific competition. Understanding this interaction is important for understanding species ecology and promoting coexistence between carnivores and people in a human-dominated landscape.

## Supporting information

**S1 Fig. Correlation between predictive variables such as farmland area (km$^2$), grassland area (km$^2$), cc: canopy cover (%), Large-predators: presence of large predators, Human: number of human presences, and Building: nearest distance to settlement (m), water: nearest distance to water body (m), LIC: Large Indian Civet, SIC: Small Indian Civet.**
(TIF)

## Acknowledgments

We thank the Department of Forests and Soil Conservation (DOFS), Department of National Parks and Wildlife Conservation (DNPWC), Parsa National Park (PNP), and Koshi Tappu Wildlife Reserve (KTWR) for providing research permission. We thank Basudha Rawal, Chandu Paneru, Sagar Parajuli, Surya Devkota and Bashu Dev Baral for data management, chief warden of PNP and KTWR and all the Division Forest Officers (DFOs) from Madhesh Province for their continuous support for this study.

## Author contributions

**Conceptualization:** Bishal Subedi, Hari Prasad Sharma.

**Data curation:** Sandeep Regmi.

**Formal analysis:** Bishal Subedi, Sandeep Regmi, Hari Prasad Sharma.

**Funding acquisition:** Hari Prasad Sharma, Bishnu Prasad Bhatarai, Hem Bahadur Kauwal.

**Investigation:** Bishal Subedi, Sandeep Regmi, Amrit Nepali, Niraj Regmi, Amir Basnet, Krishna Tamang, Bishnu Aryal, Sabin KC, Pradip Kandel, Shivish Bhandari, Bishnu Prasad Bhattarai, Hem Bahadur Katuwal, Hari Prasad Sharma.

**Methodology:** Bishal Subedi, Hari Prasad Sharma.

**Project administration:** Hari Prasad Sharma.

**Supervision:** Hari Prasad Sharma.

**Validation:** Hari Prasad Sharma.

**Visualization:** Hari Prasad Sharma.

**Writing – original draft:** Bishal Subedi, Hari Prasad Sharma.

**Writing – review & editing:** Bishal Subedi, Sandeep Regmi, Amrit Nepali, Niraj Regmi, Amir Basnet, Krishna Tamang, Bishnu Aryal, Sabin KC, Pradip Kandel, Shivish Bhandari, Bishnu Prasad Bhattarai, Hem Bahadur Katuwal, Jerrold L Belant, Ashok Kumar Ram, Hari Prasad Sharma.

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
