## [Decision Letter · Decision Letter 0]

PONE-D-24-39423Factors affecting spatio-temporal occurrence of sympatric civet in Parsa-Koshi Complex, NepalPLOS ONE

Dear Dr. Sharma,

Thank you for submitting your manuscript to PLOS ONE. After careful consideration, we feel that it has merit but does not fully meet PLOS ONE’s publication criteria as it currently stands. Therefore, we invite you to submit a revised version of the manuscript that addresses the points raised during the review process.

We look forward to receiving your revised manuscript.

Kind regards,

Taslima Sheikh

Academic Editor

PLOS ONE

Journal Requirements:

2. Thank you for stating the following financial disclosure: Tribhuvan University National Priority Area Research Grant (TU-NPAR-2078/79-ERG-04) of Tribhuvan University, Nepal

3. In the online submission form, you indicated that the data are available from corresponding author

Reviewers' comments:

Reviewer's Responses to Questions

**Comments to the Author**

1. Is the manuscript technically sound, and do the data support the conclusions?

Reviewer #1: Partly

Reviewer #2: Yes

2. Has the statistical analysis been performed appropriately and rigorously? 

Reviewer #1: No

Reviewer #2: Yes

3. Have the authors made all data underlying the findings in their manuscript fully available?

Reviewer #1: Yes

Reviewer #2: No

4. Is the manuscript presented in an intelligible fashion and written in standard English?

Reviewer #1: Yes

Reviewer #2: Yes

5. Review Comments to the Author

Reviewer #1: This study looks at how two sympatric civets occur spatially and temporally in the Parsa-Koshi Complex. While this is an interesting dataset with key questions on spatial-temporal interactions of species, there is still changes to be done before this can be considered for publication:

Abstract: You point out in your abstract that your results can have importance for conservation. I fail to see the link. Can you be more specific?

Introduction: I would like the authors to start the introduction a bit more broadly. Can you bring in existing theories on species interactions and what we know about it and then go into specific examples? From this let potential prediction emerge that you can testing. Otherwise, the important why you choose two civets remains unclear.

Line 52-60: much of this information can be in the material and methods section

General introduction point: Given that you allude to conservation in the abstract, it will be important to illustrate examples in the introduction where such information has come to conservation use.

Line 86-87: but why are you predicting this?

Line 106: Rather than just stating the reference, it would be nice to have a few sentences explaining the study design and then referring to the studies here

Line 110: how do you know that this was truly random?

Line 115-116: If the cameras were operational of 21 days how was the sampling done between Dec 2022-Mar 2023, did you keep shifting cameras? Need more clarity

Line 117-118: Was this done during field work or using other data from satellite etc?

Line 131-132: Why did you go the dredge way? This can lead to all sort of ecologically not-meaningful models being tested. A better approach would be to have a good candidate set of covariates and then build models from that.

Analysis: A table is needed to be to added which list the various covariates used and what their expected relationship with your output of interest is.

Line 138: Need to justify why the Pianka index was used and not other approaches like SIF.

Analysis: I think a bit of reanalysis might be useful here. Rather than individually looking at models for the two civets and then checking for overlap, it would be better to do a combined model (eg. the Richmond model or the Rota model) and see how the presence of one of them impacts the other, along with the covariates). This will change your analysis a bit and also your results.

Discussion: You need to broaden the scope of the discussion to not only focus on Civets in your fieldsite but to other such meso-carnivore interactions across various other sites. How similar or dissimilar are your results? What can you contribute to the larger literature on the species interactions?

Best of luck with the revisions!

Reviewer #2: A simple, well written and to the point, study. I have very little to make in the way of comments. The data supports the conclusions, the methodology was clear and concise. I perhaps would have expanded on the implications of the research to civet conservation, in terms of habitat management and probably commented on the potential issues arising for larger carnivores.

Specific minor revisions:

Line 113: the word 'trial', I am guessing should be 'trail'.

Line 113: How did they verify the animal capture area? A few details here would be useful.

Figures 3 and 4 - the axis are a little confusing in that in figure 3 large predator occurrence probability is on the x-axis whereas in figure 4 it's on the y-axis. I suggest consistency and keep the variable large predator occurrence on the x-axis, otherwise at a glance it is rather confusing.

A table containing the measured habitat variables used in the models would be helpful for a quick reference (perhaps including correlation coefficients?).

Finally, the author's state that the data is freely available from the lead author, which is fine, but placing the data in a repository would be in line with PLOS data policy.

6. PLOS authors have the option to publish the peer review history of their article (what does this mean? ). If published, this will include your full peer review and any attached files.

**Do you want your identity to be public for this peer review?** For information about this choice, including consent withdrawal, please see our Privacy Policy .

Reviewer #1: **Yes: ** Munib Khanyari

Reviewer #2: No

---

## [Author Response · Author response to Decision Letter 1]

12 May 2025

Dear Editor

Plos One

We highly appreciate the constructive comments and edits provided by reviewers and editor to shape our manuscript. Please find our responses in italic with yellow color, and we also marked the changes text with yellow color in the manuscript.

Comments from editor

Removal of Funding Information from the Manuscript:

Response: At your suggestion, we removed funding information from the manuscript, as per journals guidelines.

2). Data Availability Statement

Response: All relevant data are within the manuscript.

3). Image License and Protected Area Shape File Clarification:

Response: We apologize for the confusion regarding the map's attribution and license information in Figure 1. Upon further review, we have now removed the Land Use and Land Cover map. As suggested, we have remade the map using OpenStreetMap data and included the following attribution in the figure 1 caption. "Contains information from OpenStreetMap and OpenStreetMap Foundation, which is made available under the Open Database License." With appropriate citation " OpenStreetMap Contributors. Planet dump [Data file from 2023]. (2023) https://planet.openstreetmap.org." in number 35 in line 129-130.

Reviewer #1: This study looks at how two sympatric civets occur spatially and temporally in the Parsa-Koshi Complex. While this is an interesting dataset with key questions on spatial-temporal interactions of species, there is still changes to be done before this can be considered for publication:

Abstract: You point out in your abstract that your results can have importance for conservation. I fail to see the link. Can you be more specific?

Response: We highly appreciate your suggestion, and revised the abstract to inform conservation efforts, particularly in designing site-specific strategies that account for habitat preferences and predator impacts on carnivore populations (Lines 27-36).

Introduction: I would like the authors to start the introduction a bit more broadly. Can you bring in existing theories on species interactions and what we know about it and then go into specific examples? From this let potential prediction emerge that you can testing. Otherwise, the important why you choose two civets remains unclear.

Response: At your suggestion, we revised the introduction to begin with borders theories on species interactions, such as niche differentiation and resource partitioning. We than added specific examples to contextualize these theories and clarify the rationale for selecting the two civet species, leading to our predictions and study objectives. The changes have been made throughout the introduction section.

Line 52-60: much of this information can be in the material and methods section.

General introduction point: Given that you allude to conservation in the abstract, it will be important to illustrate examples in the introduction where such information has come to conservation use.

Response: At your recommendation, we moved the detailed descriptions of civet habitat preferences, distribution, and environmental variables (Lines 52–60), to the Materials and Methods section to streamline the introduction and ensure the appropriate placement of this information.

Line 86-87: but why are you predicting this?

Response: We rephrased the sentence to be more informative.

Line 106: Rather than just stating they p reference, it would be nice to

have a few sentences explaining the study design and then referring to

the studies here.

Response: Our apology for misunderstanding, the information is on study design (Lines 105-109).

Line 110: how do you know that this was truly random?

Response: We have now changed the information in the line 108 to make it more informative and correct.

Line 115-116: If the cameras were operational of 21 days how was the

sampling done between Dec 2022-Mar 2023, did you keep shifting cameras?

Need more clarity.

Response: We deployed 154 cameras sequentially across different grid cells during the study period, with each set operational for 21 days. Once cameras completed their deployment in one grid cell, they were shifted to new locations in subsequent grid cells, ensuring coverage across the study area. This approach allowed efficient use of available equipment while covering the spatial extent of the Parsa-Koshi Complex. We have added this information in Lines 114-116

Line 117-118: Was this done during field work or using other data from

satellite etc.?

Response: The measurement of canopy cover was conducted during fieldwork using the Gap Light Analysis Mobile Application (GLAMA) at each camera station. The area of grassland within a 2500 m radius of each camera station was determined using satellite-derived ESRI Sentinel-2 land-use land-cover maps with a 10 m resolution. This map source is open access and made available under a Creative Commons Attribution 4.0 (CC BY 4.0) license. This means people are free to use, share, and adapt the data as long as you provide proper attribution to the data providers (Impact Observatory, Esri, and Microsoft). The correct citation for this map source is “Impact Observatory, Esri, & Microsoft. (2022). Sentinel-2 10m Land Cover Explorer [Data set: 45R_20230101-20240101]. Esri. Retrieved from https://livingatlas.arcgis.com/landcover”. Also mentioned at https://www.impactobservatory.com/static/c033eb846160f6a0a35c63a64ef45e52/lulc-methodology-accuracy.pdf .These two complementary methods ensured accurate characterization of habitat variables.

Line 131-132: Why did you go the dredge way? This can lead to all sort

of ecologically not-meaningful models being tested. A better approach

would be to have a good candidate set of covariates and then build

models from that.

Response: We highly appreciate your suggestion. We used the dredge approach due to the exploratory nature of our analysis, aiming to identify key predictors without bias from predefined covariates. To ensure ecological relevance, we validated top models against existing literature and focused on variables supported by both statistical and ecological evidence.

Analysis: A table is needed to be to added which list the various

covariates used and what their expected relationship with your output of

interest is.

Response: We have included all covariates included in table and these are described in the table caption.

Line 138: Need to justify why the Pianka index was used and no other approaches like SIF.

Response: We chose the Pianka index because it is a widely used and robust method for quantifying niche overlap based on frequency of detections. While other indices, such as SIF, could be employed, the Pianka index is well-suited to our data, where presence-absence from camera traps was used. We have also clarified this choice in the revised manuscript.

Analysis: I think a bit of reanalysis might be useful here. Rather than

individually looking at models for the two civets and then checking for

overlap, it would be better to do a combined model (eg. the Richmond

model or the Rota model) and see how the presence of one of them impacts

the other, along with the covariates). This will change your analysis a

bit and also your results.

Response: We highly appreciate your critical suggestion to explore a combined modeling approach, such as the Richmond or Rota model, to assess how the presence of one species impacts the other. While this method provides valuable insights, we chose to analyze the occurrence of Large Indian Civet (LIC) and Small Indian Civet (SIC) separately, followed by evaluating their overlap, to align with the approach used in comparable studies. For example, Akrim et al., 2023, employed a similar framework to assess spatial and dietary overlap between Small Indian Civet and Asian Palm Civet in Pakistan. This methodology has been demonstrated to effectively identify occurrence patterns and niche overlaps between sympatric species without requiring combined models.

Additionally, our focus was on understanding the environmental and ecological variables driving individual species' occurrences while evaluating overlap patterns post hoc to infer their interactions. We acknowledge the importance of combined models in species interaction studies and agree that this would be a valuable avenue for future research to build on our findings. However, for this study, we believe our current approach is appropriate and scientifically robust for addressing the research objectives. After your suggestion we tried to fit two species interaction approach following Richmond and Rota models however this produced insignificant results.

Discussion: You need to broaden the scope of the discussion to not only

focus on Civets in your field site but to other such meso-carnivore

interactions across various other sites. How similar or dissimilar are

your results? What can you contribute to the larger literature on the

species interactions?

Response: At your suggestion, we revised the discussion to place our findings within a broader context of meso-carnivore interactions globally. By comparing our results with similar studies, we highlight their contribution to the understanding of niche partitioning and coexistence strategies in sympatric carnivores.

Reviewer #2: A simple, well written and to the point, study. I have very

little to make in the way of comments. The data supports the

conclusions, the methodology was clear and concise. I perhaps would have

expanded on the implications of the research to civet conservation, in

terms of habitat management and probably commented on the potential

issues arising for larger carnivores.

Response: Thank you for critical comments and suggestions. We have addressed all your comments and suggestions in the revised manuscript.

Specific minor revisions:

Line 113: the word 'trial', I am guessing should be 'trail'.

Response: At your suggestion, we have changed it.

Line 113: How did they verify the animal capture area? A few details here would be useful.

Response: We highly appreciate your suggestion, we ensured optimal camera placement by inspecting trails for animal signs (e.g., tracks, scats), adjusting the camera angle and height accordingly, and conducting test shots to confirm full coverage of the target area. We have now modified the phrase to make it easier to understand.

Figures 3 and 4 - the axis are a little confusing in that in figure 3 large predator occurrence probability is on the x-axis whereas in figure 4 it's on the y-axis. I suggest consistency and keep the variable large predator occurrence on the x-axis, otherwise at a glance it is rather confusing.

Response: At your suggestion, we have corrected the inconsistency in the revised manuscript

A table containing the measured habitat variables used in the models

would be helpful for a quick reference (perhaps including correlation

coefficients?).

Response: We highly appreciate your concern, and it is included in a Supplementary Figure 1.

---

## [Editor Report · Decision Letter 1]

Factors affecting spatio-temporal occurrence of sympatric civets in Parsa-Koshi Complex, Nepal

PONE-D-24-39423R1

Dear Dr. Sharma

We’re pleased to inform you that your manuscript has been judged scientifically suitable for publication and will be formally accepted for publication once it meets all outstanding technical requirements.

Kind regards,

Taslima Sheikh

Academic Editor

PLOS ONE
---

## [Editor Report · Acceptance letter]

PONE-D-24-39423R1

PLOS ONE

Dear Dr. Sharma,

I'm pleased to inform you that your manuscript has been deemed suitable for publication in PLOS ONE. Congratulations! Your manuscript is now being handed over to our production team.

Kind regards,

on behalf of

Dr. Taslima Sheikh

Academic Editor

PLOS ONE